# Selective experience replay compression using coresets for lifelong deep reinforcement learning in medical imaging

**Guangyao Zheng**[1]                            TZ30@RICE.EDU
**Samson Zhou**[1]                        SAMSONZHOU@GMAIL.COM
**Vladimir Braverman**[1]                        VB21@RICE.EDU
**Michael A. Jacobs**[2,3]              MICHAEL.A.JACOBS@UTH.TMC.EDU
**Vishwa S. Parekh**[4]            VPAREKH@SOM.UMARYLAND.EDU

[1] *Department of Computer Science, Rice University, Houston, TX, USA*

[2] *Department Of Diagnostic And Interventional Imaging, McGovern Medical School, UTHealth Houston, Houston, TX, USA,* [3] *The Russell H. Morgan Department of Radiology and Radiological Science, The Johns Hopkins University School of Medicine, Baltimore, MD 21205*

[4] *University of Maryland Medical Intelligent Imaging (UM2ii) Center*

*Department of Diagnostic Radiology and Nuclear Medicine*

*University of Maryland School of Medicine*

*Baltimore, MD 21201*

**Editors:** Accepted for publication at MIDL 2023

## Abstract

Selective experience replay is a popular strategy for integrating lifelong learning with deep reinforcement learning. Selective experience replay aims to recount selected experiences from previous tasks to avoid catastrophic forgetting. Furthermore, selective experience replay based techniques are model agnostic and allow experiences to be shared across different models. However, storing experiences from all previous tasks make lifelong learning using selective experience replay computationally very expensive and impractical as the number of tasks increase. To that end, we propose a reward distribution-preserving coreset compression technique for compressing experience replay buffers stored for selective experience replay.

We evaluated the coreset lifelong deep reinforcement learning technique on the brain tumor segmentation (BRATS) dataset for the task of ventricle localization and on the whole-body MRI for localization of left knee cap, left kidney, right trochanter, left lung, and spleen. The coreset lifelong learning models trained on a sequence of 10 different brain MR imaging environments demonstrated excellent performance localizing the ventricle with a mean pixel error distance of 12.93, 13.46, 17.75, and 18.55 for the compression ratios of 10x, 20x, 30x, and 40x, respectively. In comparison, the conventional lifelong learning model localized the ventricle with a mean pixel distance of 10.87. Similarly, the coreset lifelong learning models trained on whole-body MRI demonstrated no significant difference (p=0.28) between the 10x compressed coreset lifelong learning models and conventional lifelong learning models for all the landmarks. The mean pixel distance for the 10x compressed models across all the landmarks was 25.30, compared to 19.24 for the conventional lifelong learning models. Our results demonstrate that the potential of the coreset-based ERB compression method for compressing experiences without a significant drop in performance.

**Keywords:** Deep reinforcement learning, lifelong learning, continual learning, medical imaging, coresets, clustering

## 1. Introduction

The field of radiology is moving towards implementation of artificial intelligence (AI) methods for radiologists to use in advanced reading rooms. Deep reinforcement learning (DRL) is an evolving area of research within the field of AI that deals with the development of AI systems that can learn from experience (mimicking the human learning process). DRL has produced excellent results across diverse domains (Mnih et al., 2013; Li et al., 2016; Silver et al., 2017; Sallab et al., 2017).

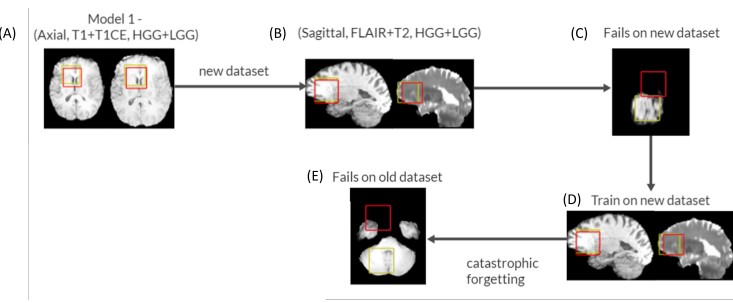

Figure 1: Illustration of catastrophic forgetting in dynamically evolving medical imaging environments.

The property of learning from experience makes DRL algorithms ideal for deployment into radiological decision support systems, where the DRL models can learn how to map the intra- and inter-structural relationships within different radiological images. The use of DRL in radiological applications is an emerging area of active research with new techniques being developed for anatomical landmark localization, image segmentation, registration, treatment planning, and assessment (Ghesu et al., 2017; Tseng et al., 2017; Maicas et al., 2017; Ma et al., 2017; Alansary et al., 2018; Ali et al., 2018; Alansary et al., 2019; Vlontzos et al., 2019; Allioui et al., 2022; Zhang et al., 2021; Joseph Nathaniel Stember, 2022).

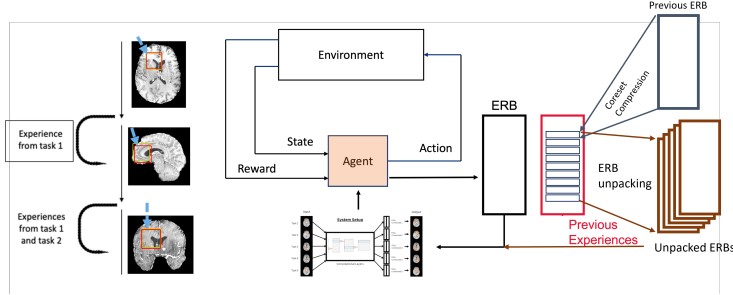

Figure 2: A schematic of the coreset-compressed lifelong deep reinforcement learning setup for training deep reinforcement learning models. ERB=Experience Replay Buffer

In medical imaging applications, the same task might be present within different radiological imaging environments. For example, brain imaging will involve different imaging orientations (axial, sagittal, or coronal), different imaging sequences (ie, MRI: T1, T2, FLAIR, PWI), and modalities (PET, CT) or different pathologies (low-grade or high-grade gliomas), potentially resulting in large environments, depending on the application. Furthermore, the imaging environments for medical imaging tasks are constantly evolving, i.e., newer environments might be present at future time points due to change in image acquisition parameters or the introduction of

newer imaging sequences. As a result, a model trained in an old environment may fail when evaluated in a new unseen environment. The model could then be fine-tuned using different learning methods to work in the new environment, However, these methods could potentially result in catastrophic forgetting, where the model is no longer capable of operating in the original environment. Such an example is shown in Figure 1.

The challenges of catastrophic forgetting could be addressed using lifelong deep reinforcement learning. Elastic weight consolidation (EWC) (Kirkpatrick et al., 2017) and selective experience replay (Rolnick et al., 2019) are two commonly used techniques for integrating lifelong learning with reinforcement learning. Elastic weight consolidation aims to preserve the network parameters learned in the previous tasks while learning a new task. On the other hand, selective experience replay aims to recount selected experiences from previous tasks to avoid catastrophic forgetting. Furthermore, selective experience replay based techniques are model agnostic and allow experiences to be shared across different models. However, storing experiences from all previous tasks make lifelong learning using selective experience replay computationally very expensive and impractical as the number of tasks increase. To that end, we develop a reward distribution-preserving coreset compression technique based on weighted sampling to compress experience replay buffers (ERBs) stored for lifelong learning without sacrificing the performance, as shown in Figure 2. We evaluated the proposed coreset based ERB compression technique for the task of ventricle localization in brain MRI across a sequence twenty-four different imaging environments consisting of a combination of different MRI sequences, diagnostic pathologies, and imaging orientations.

## 2. Related Work

Compression and Sampling experience replays has been an active area of research in the field continual reinforcement learning (Schaul et al., 2015; Pan et al., 2022; Ramicic et al., 2022; Hessel et al., 2018; Tiwari et al., 2021; Lazic et al., 2021). The majority of the previous work on experience replay sampling is built on top of the work by Schaul et. al. (Schaul et al., 2015), in which the authors proposed a method to use temporal difference (TD) error to prioritize the more valuable experiences. Further work explored the idea of prioritized experience replay and developed new methods that improved the results (Pan et al., 2022; Ramicic et al., 2022; Hessel et al., 2018). However, they all need an extra calculation for absolute TD error, which is constantly updated during the network's training session, since the TD error requires the Q function, and the Q function updates after every training iteration. In contrast, recent work from Tiwari et al. (Tiwari et al., 2021) used gradient coreset based experience replay sampling with excellent results. However, a major limitation of the proposed approach was the required access to the training model for compression. Alternatively, Lazic et. al. (Lazic et al., 2021) utilized q-functions for sampling a coreset from ERBs, which could potentially be a major limitation.

To that end, the goal of this work was to develop a coreset based ERB sampling technique that does not require additional information from the training session nor does it require the model parameters from the training session, thereby addressing the shortcomings of the current techniques. Furthermore, the proposed technique only requires the experience replay buffer for compression. This is an asynchronous process and can be used to learn

from reinforcement learning models that just use an experience replay buffer, but do not have any additional implementation.

## 3. Methods

### 3.1. Lifelong deep reinforcement learning

We implemented a deep learning framework based on the deep Q-network (DQN) algorithm, as illustrated in Figure 2. The 3D DQN model implemented in this work was adapted from (Mnih et al., 2013; Alansary et al., 2018; Vlontzos et al., 2019; Parekh et al., 2020). The environment was represented by the 3D imaging volume and the agent, by a 3D bounding box with six possible actions, $a \in \{$x++, x--, y++, y--, z++, z--$\}$. The state is defined by the current location (or a sequence of locations), where each location is represented by a 3D bounding box. The reward is defined by the difference in the distance between the agent's location and the target landmark before and after the agent's action. The state-action-reward-resulting state $[s, a, r, s']$ tuples resulting from the DRL agent's interaction with the environment over many episodes were used to populate the experience replay buffer (ERB).

To perform lifelong learning, we implemented a selective experience replay buffer to collect a trajectory of experience samples across the model's training history. The model attempts to learn a generalized representation of its current and previous tasks by sampling a batch of experience from both its current task's experience replay buffer (ERB) as well as from its history of previous tasks' experience replays during training.

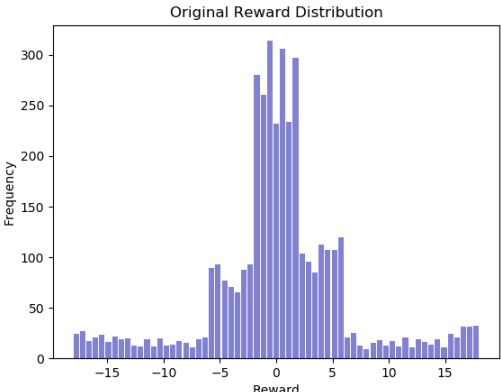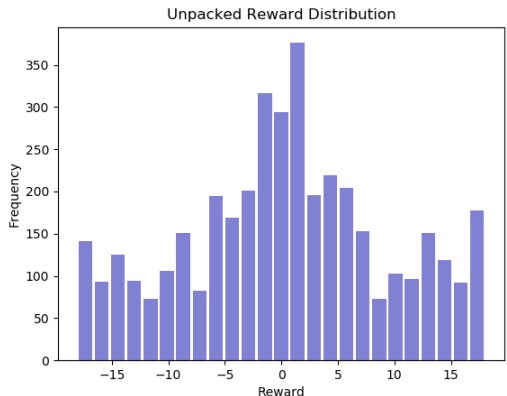

Figure 3: Reward distribution of original ERB (left) vs. compressed and unpacked ERB with a compression ratio of 10x (right)

### 3.2. Weighted sampling-based reward distribution preserving coreset ERB compression

We proposed and introduce a new method to compress the ERB based on coresets for lower storage cost, less communication cost, and preservation of information: given an ERB of size $N$ and compression rate of $R$, we use $k$-means++ clustering to partition the points in

the ERB to $N/R$ clusters based on their rewards. We use $k$-means++ clustering because we want to manually pick the number of clusters to be $N/R$ as it reflects the compression ratio. Moreover, since the reward distribution for ERBs is known, does not have outliers, and is only 1-dimensional, $k$-means++ clustering can guarantee convergence on a good label assignment very fast and accurately. We pick the closest point to the center of these clusters to be in the compressed ERB. Additionally, we give these points in the compressed ERB a weight parameter that is equal to the number of points in the clusters they are in. When the compressed ERB is received by an agent, the agent will unpack it by repeating the points in the compressed ERB multiple times according to their weight parameter. For example, if a point $[s, a, r, s']$ has a weight of 2 in the compressed ERB, then it will be repeated twice in the unpacked ERB. This method guarantees an approximate reward distribution similar to the original reward distribution, as illustrated in Figure 3.

## 4. Experiment and Data

### 4.1. Experiment 1: Brain MRI

#### 4.1.1. Clinical Data

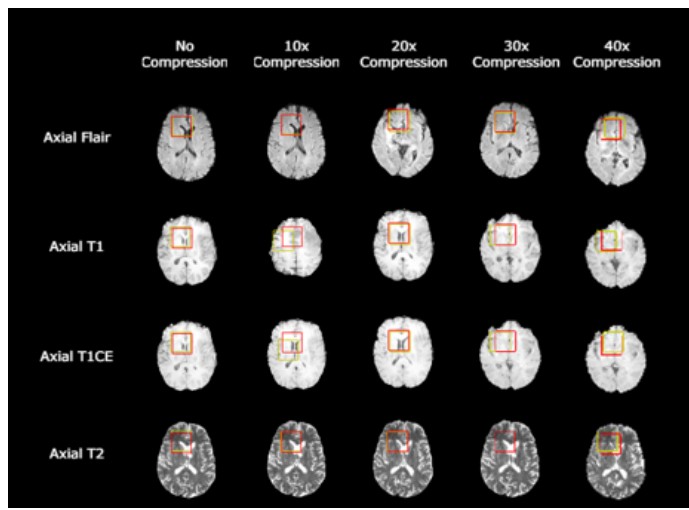

Figure 4: Illustration of a subset of task-environment pairs in the BRATS dataset and the performance of no compression and the proposed coreset-compression method at different compression rates. The red and yellow bounding boxes represent the ground truth and the agent's prediction, respectively.

We used the brain tumor segmentation (BRATS) dataset for evaluating the coreset compression method (Menze et al., 2014). The dataset consisted of 285 patients with T1-weighted pre- and post-contrast enhanced, T2-weighted, and Fluid Attenuated Inversion Recovery (FLAIR) sequences. All the images were acquired in the axial orientation. In this work, we selected a random subset of 100 patients from the BRATS dataset for the development, training, and evaluation of different DRL models. This subset consisted of 60 patients with high-grade glioma (HGG) and 40 patients with low-grade glioma (LGG). Of the 100 patients, 80 were used for training and 20 were used for evaluation. The training dataset consisted of 48 patients with HGG and 32 patients with LGG tumors. The test dataset consisted of 12 patients with HGG and 8 patients with LGG tumors. We synthetically resliced the axial images into coronal and sagittal views and reconstructed each dataset in all three imag-

ing orientations, resulting in a total of twenty-four imaging environments (2pathologies × 4sequences × 3orientations = 24). We used the top left ventricle as the task for this experiment, resulting in a total of 24 task-environment pairs, as shown in Figure 4.

### 4.1.2. TRAINING PROTOCOL

We trained 5 lifelong deep reinforcement learning models (no-compression, 10x compression, 20x compression, 30x compression, and 40x compression) to test the performance of our coreset compression algorithm. The models are trained for the localization of the top left ventricle in a randomly sampled 10 task-environment pairs out of the 24 task-environment pairs. The models were trained for four epochs with a batch size of 48. The agent's state was represented as a bounding box of size 45x45x11 with a frame history length of four. The models were iteratively trained for the localization of the top left ventricle in one imaging environment at a time, resulting in 10 rounds of training. The no-compression model used complete ERBs generated by previous training rounds for the next training round. The 10x, 20x, 30x, and 40x compression models used 10x, 20x, 30x, and 40x compressed ERBs, respectively, and unpacked based on weights in the ERBs from all previous rounds for the next training round.

## 4.2. Experiment 2: Whole Body MRI

### 4.2.1. CLINICAL DATA

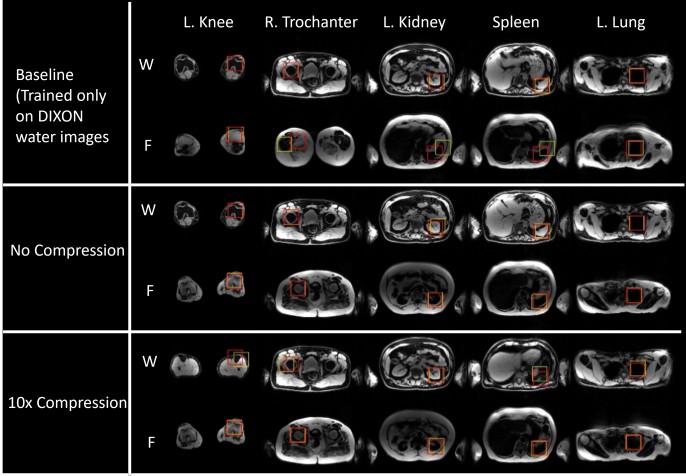

Figure 5: Illustration of the performance of no-compression and coreset compression lifelong learning models on the two imaging environments and five tasks for whole body MRI.

The WB multiparametric MRI data set consisted of thirty subjects acquired using the imaging protocol that scanned from the shoulders to the lower mid calf and described in (Leung et al., 2020). We evaluated the proposed coreset ERB compression technique for training lifelong reinforcement learning models to localize five landmarks (left lung, left kidney, right trochanter, spleen, and the left knee cap) across two imaging environments (DIXON Fat and DIXON water images) in this study, as shown in Figure 5

### 4.2.2. TRAINING PROTOCOL

We trained two lifelong deep reinforcement learning models (no-compression and 10x compression) to test the performance of our coreset compression algorithm.

The dataset comprising of 30 subjects was randomly divided into three groups - ten for training, ten for lifelong learning, and the remaining ten for testing. Furthermore, the first group comprised of only the DIXON water and the second group of only the DIXON fat imaging sequences. The models were iteratively trained (using the same hyperparameters as the first experiment) for the localization of each of the five landmars in one imaging environment at a time, resulting in two rounds of training.

### 4.3. Performance Evaluation

The performance metric was set as the terminal Euclidean distance between the agent's prediction and the target landmark. We performed paired t-tests to compare the performance of the no-compression model with the compression models at different compression rates. The p-value for statistical significance was set to $p \leq 0.05$.

## 5. Results

For the first experiment on the BRATS dataset, we sequentially trained each selective experience replay based lifelong reinforcement model (no compression, 10x compression, 20x compression, 30x compression, and 40x compression) on 10 distinct task-environment pairs, one pair each round. After every round of training, the models were evaluated across all 24 task-environment pairs in the test set. Figure 4 illustrates the performance of all five models across a subset of four task-environment pairs after 10 rounds of training. As shown in Figure 4, both the compressed and uncompressed models demonstrated excellent continual learning performance across all task-environment pairs. The average Euclidean distance errors for no compression, 10x compression, 20x compression, 30x compression, and 40x compression were 10.87, 12.93 ($p = 0.01$), 13.46 ($p = 0.0001$), 17.75 ($p \leq 0.0001$), and 18.55 ($p \leq 0.0001$) respectively. Figure 6 (left) compares the performance at different compression levels across different rounds. The performance after 10 rounds of training across different compression levels has been illustrated as a box plot in Figure 6 (right). The original sizes of the ERBs tested were 90MB. The 10x, 20x, 30x and 40x coreset compression resulted in ERBs of size 9MB, 4.5MB, 3MB, and 2MB, respectively.

For the second experiment on the whole-body MRI dataset, we sequentially trained each selective experience replay based lifelong reinforcement model (no compression and 10x compression) on two distinct environments (one each round) across all five landmarks. After every round of training, the models were evaluated across both the environments pairs in the test set. Figure 5 compares the performance the baseline (trained only on the DIXON water images), no-compression, and 10x compression models across all five landmarks for an example patient. Table 1 summarizes the pixel distances demonstrating no significant difference between no compression and 10x compression models. The coreset technique here compressed the ERBs from 750MB to 75MB.

## 6. Discussion

In conclusion, we propose a coreset-based ERB compression technique for increased computational efficiency and scalability of selective experience replay based deep lifelong reinforcement learning with excellent performance. In particular, our weighted sampling-based

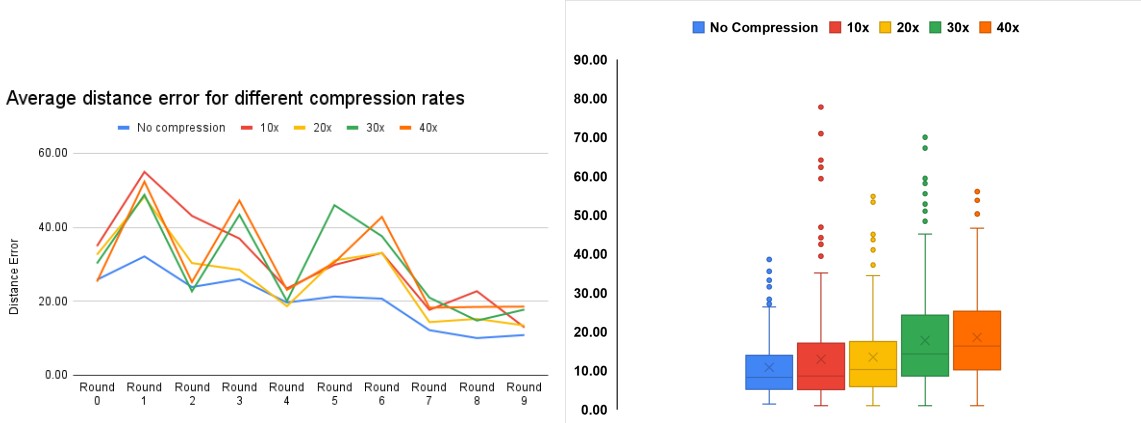

Figure 6: **Left**: Average Euclidean distance error on testing images (24 task-environment pairs) after every training round. **Right**: Box plot of the Euclidean distance error on testing images (24 task-environment pairs) after ten rounds of training.

Table 1: Comparison between conventional Lifelong Learning and Coreset Lifelong Learning for the localization of five landmarks across two imaging environments in whole body MRI

|  | knee | trochanter | kidney | lung | spleen | Average |
|---|---|---|---|---|---|---|
| Baseline (trained on just DIXON water) | 73.44 | 45.05 | 100.33 | 40.25 | 20.63 | 55.94 |
| No compression | 6.34 | 30.95 | 25.50 | 21.71 | 11.69 | 19.24 |
| Coreset 10x compression | 15.74 | 19.93 | 36.97 | 35.92 | 17.92 | 25.30 |
| Paired TTEST (coreset vs. conventional LL) | 0.27 | 0.65 | 0.15 | 0.06 | 0.13 | 0.28 |

reward distribution-preserving coreset ERB compression showed excellent performance for shrinking the size of saved experiences from previous tasks for lifelong deep reinforcement learning in medical imaging. Our results demonstrated that experience replay buffers can be compressed up to 10x without any significant drop in performance.

Experience replay has been one of the major methods for lifelong reinforcement learning because it is model agnostic and because different experiences can be shared between models for collaborative lifelong learning. However, the computational complexity of storing multiple experiences from previous tasks makes experience replay less attractive as the number of tasks increases. Many techniques have been developed in the literature for sampling experience replay (Schaul et al., 2015; Pan et al., 2022; Ramicic et al., 2022; Hessel et al., 2018). However, they required additional calculations that constantly update during the network's training and require the Q function. In contrast, our proposed coreset-based compression techniques asynchronously compress the ERB, without the need for extra information from the training session where the ERB is produced or the model parameters that came from the training session. This provides more flexibility to the training and is less computationally intensive compared to other methods.

There are certain limitations to our study. This work approximates the joint distribution of state-action-reward-next state using the reward distribution alone, resulting in a potential

loss of information. For example, two state-action-reward-next state tuples with the same reward may have different states, and our method would only pick one of the state-action-reward-next state tuple. We plan to address this limitation by incorporating state and action into the distribution preserving coreset compression method in the future. The second limitation of this study is the limited focus on single-agent deep reinforcement learning models. In the future, we plan to evaluate the coreset-based ERB compression technique in a multi-agent setup where the size of the ERB linearly increases with the number of agents and across a diverse set of applications.

## Acknowledgments

Funding: This work was supported by the DARPA grant: DARPA-PA-20-02-11-HR00112190130 and 5P30CA006973 (Imaging Response Assessment Team-IRAT), U01CA140204

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

## Appendix A. Background on Coresets

Coresets are an important concept in data science because they present a technique for representing datasets with a large number of observations as weighted datasets with a smaller number of observations. Coresets are frequently used as a pre-processing technique for dimensionality reduction to significantly improve the runtime or memory cost of down-streams tasks, such as clustering, regression, or low-rank approximation. See the surveys (Feldman, 2020; Bachem et al., 2017) for a more comprehensive introduction and discussion on coresets, their applications, and the state-of-the-art techniques.

More formally, we are given a dataset $X = x_1, \ldots, x_n$ such that $x_i \in \mathbb{R}^d$ for all $i \in [n]$. In other words, the dataset $X$ consists of $n$ observations, each with $d$ features. The goal is to minimize the function $f(X, q, u)$ across all possible values $q$ in some query space $Q$. Here we use $u$ to denote the function that assigns unit weight to all of the $n$ points in $X$. For example, if the task was to do $k$-means clustering, then $Q$ would be the set consisting of all sets of $k$ points from $\mathbb{R}^d$, i.e., the set of all possible sets of centers for the clustering problem.

A coreset construction produces a weighted subset $Y$ consisting of $m$ observations from the original dataset $X$. Let $w$ be the function that assigns weights to points in $Y$. Given an approximation parameter $\epsilon > 0$, the coreset with multiplicative error $(1 + \epsilon)$ satisfies

$$(1 - \epsilon)f(Y, q, w) \leq f(X, q, u) \leq (1 + \epsilon)f(Y, q, w),$$

for all $q \in Q$. Similarly, a coreset with additive error $E > 0$ satisfies

$$|f(Y, q, w) - f(X, q, u)| \leq E.$$

Since the coreset is defined with respect to $f$, then it naturally follows that there may be drastically different coreset constructions depending on the task at hand, i.e., the function $f$ to be minimized.

The definition can then be naturally generalized to the setting where $X$ is a set of weighted points. Formally, we have the following:

**Definition 1 (Coreset)** *Given a set $X$ with weight function $u$ and an accuracy parameter $\epsilon > 0$, we say a set $Y$ with weight function $w$ is an $(1 + \epsilon)$-multiplicative coreset for a function $f$, if for all queries $q$ in a query space $Q$, we have*

$$(1 - \epsilon)f(Y, q, w) \leq f(X, q, u) \leq (1 + \epsilon)f(Y, q, w).$$

A standard approach in coreset constructions is to first assign a *sensitivity* $s(x_i)$ to each point $x_i$ with $i \in [n]$. The sensitivity is intuitively a value that quantifies the "importance" of the point $x_i$. The coreset construction then samples a fixed number $m$ of points from $X$, with probabilities proportional to their sensitivity. That is, for each of the $m$ sampled points $p$, we have that $\Pr[p = x_i] \propto s_i$. In fact, the following statement shows this procedure can generate a coreset even if we only have approximations to the sensitivities of each point for the task of $k$-means clustering:

**Theorem 2 (Theorem 35 in (Feldman et al., 2020))** *Let $C > 1$ be a universal constant and for each $i \in [n]$, let $q(x_i)$ be a $C$-approximation to the sensitivity $s(x_i)$ for any*

point $x_i$. Let $T = \sum_{i=1}^{n} q(x_i)$. Then sensitivity sampling $m = O\left(\frac{Tk}{\epsilon^2} \log^2 k\right)$ points with replacement, i.e., choosing each of the $m$ points to be $x_i$ with probability proportional to $q(x_i)$ and then rescaling by the sampling probability, outputs a $(1 + \epsilon)$-coreset for $k$-means clustering with probability $\frac{2}{3}$.

However, it turns out that even approximating the sensitivities of each point may be time-consuming. Thus for real-world datasets, it is often more practical to follow procedures that capture the main intuition behind sensitivity sampling without actually performing sensitivity sampling.

**Uniform sampling.** One reason to assign different sensitivities to each observation is the following. Suppose there exist $n - 1$ similar observations and a single drastically different observation. It may be possible the drastically different observation is an outlier that should be ignored, but it may also be possible that the different observation is a crucial but rare counterexample that behaves differently from the rest of the population, in which case the counterexample needs to be sampled into the coreset in order to maintain representation. Thus in this case, the sensitivity of the different observation should be much larger than the sensitivities of the remaining points.

However, if the dataset is roughly uniform, then no particular example stands out, and so intuitively, the sensitivities of the points are also roughly uniform. In this case, the intuition behind sensitivity sampling is also achieved through uniform sampling, i.e., selecting a number of observations uniformly at random.

**Group sampling.** A frequent phenomenon is that the observations in the dataset can be clustered or partitioned into a number of groups, so that the behavior of the observations is similar within a group but drastically different across groups. In this case, the sensitivities within each group are similar, but the sensitivities across different groups can also vary greatly. Hence, sensitivity sampling would sample roughly an equal number of observations from each group.

If a natural partition of the observations into the groups is known a priori, then sensitivity sampling can be reasonably simulated by first partitioning the observations into these groups. We can then sample a fixed number of points from each group, uniformly at random. Specifically, if the coreset construction for sensitivity sampling intended to store $m$ points and $k$ groups were formed from the natural partition of the observations, then we can sample $\frac{m}{k}$ points from each of the $k$ groups. That is, for each of the $k$ groups, $\frac{m}{k}$ points are selected uniformly at random from this group and then weighted proportional to the number of points in the group.

## Appendix B. Comparison between the proposed clustering based coresets with uniform and inverse CDF sampling methods

Clustering gives us an intuition on how important the samples are by the number of samples within each cluster. Uniform sampling or inverse cdf sampling do indeed preserve the distribution, but do not provide information about the importance, resulting in just a subsample of the original data. With our method incorporating importance or "weight", we can generate the same amount of distribution preserving subsample of the original data with much smaller size. Furthermore, we do not have explicit access to the reward distribution and so

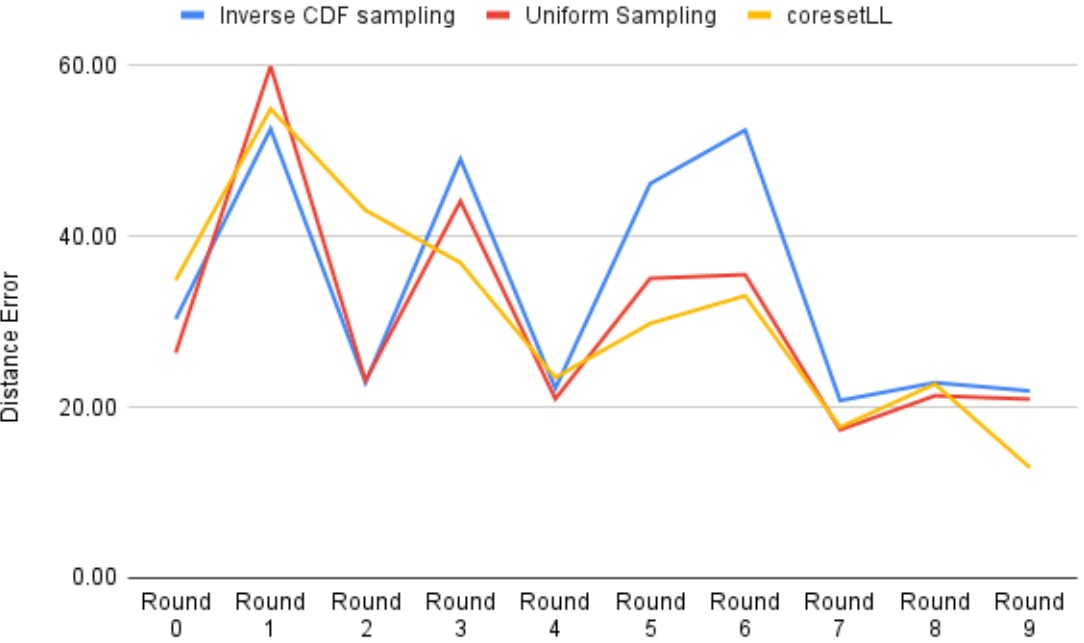

Figure 7: Comparison between the proposed clustering technique with the uniform and inverse cdf sampling techniques for 10x coreset compression for the BRATS experiment

naively performing inverse CDF sampling does not seem immediately possible. However, given a sufficiently large number of samples, it does seem possible to construct a rough estimate of the reward distribution, though this is already one possible source of error.

In addition, uniform sampling or inverse CDF sampling will provide a good representation of the original data when the original data is roughly uniformly distributed. Indeed, when the data is roughly uniformly, our clustering approach will also perform well. However, in cases where there is a small fraction of the data that performs exceptionally well or exceptionally poorly, these inputs will not be captured by uniform sampling, but may be captured by clustering.

Appendix Figure 7 demonstrates an experimental comparison between uniform sampling, cdf sampling, and the proposed clustering technique. As shown in the figure, the clustering based 10x compression achieves an average pixel distance error of 12.93, signficantly lower than uniform sampling (20.95, $p \leq 0.0001$) and inverse cdf sampling (21.91, $p \leq 0.0001$)

