# OpenReview forum: "Selective experience replay compression using coresets for lifelong deep reinforcement learning in medical imaging"
_MIDL.io/2023/Conference — MIDL 2023 Poster_

### Official Review · Reviewer_VwTz · 2023-02-04

**Confidence:** 3
**Preliminary Rating:** 4
**Recommendation:** Poster

**Summary:**

The authors propose the coreset-based Experience Replay Buffer life-long deep reinforcement learning framework for segmenting the brain tumor (BRATS) dataset for the task of ventricle localization. They develop a reward distribution-preserving coreset compression technique based on weighted sampling to compress experience replay buffers stored for lifelong learning. The method achieves 10x compression while maintaining performance comparable to the conventional method.


**Strengths:**

This work considers deep reinforcement learning for clinical deployment perspectives.

The work addresses catastrophic forgetting when adapting a model from one context to another.

The authors have considered a sequence of different imaging environments combining different MRI sequences, diagnostic pathologies, and imaging orientations.

The proposed technique is model agnostic and asynchronously compresses the experience replay buffer (ERB) without requiring extra information or model parameters.

The method is precise and less computationally intensive compared to other existing techniques.


**Weaknesses:**

The authors claim that they sample a batch of experience from current ERB and previous tasks’ unpacked ERB, but how do they sample? Do they sample it randomly with 50%-50%?

The unpacking step of the compressed ERBs is not shown in Figure 2. The authors simply merge the red and black lines to show that the previous and current ERBs are used for the DRL process.

The authors claim that they sample a batch of experience from current ERB and previous tasks’ unpacked ERB, but how do they sample? Do they sample it randomly with 50%-50%?

In Figure 4, the agent’s prediction seems to deviate from the target location for Axial T1 and Axial T1 CE, and that too for 10x compression.

The authors have not compared their methods with at least some of the other ERB sampling methods that require additional parameters like temporal difference which depends on the Q function.


**Deanonymize Review:**

no

**Detailed Comments:**

 a sequence twenty-four different imaging environments —-> should this part of the sentence be
 a sequence of twenty-four different imaging environments?

The abbreviation ERB comes on page 3 but is abbreviated multiple times on page 4 as “experience replay buffer (ERB)”. The abbreviation is found at 3 places in the manuscript which is unnecessary.

The authors claim six possible bounding box actions - up,  down, front, back, left, and right. Can more actions like diagonal movements be considered to arrive at different experiences? Can that have a better localization of the ventricle?


**Paper Type:**

methodological development

**Questions To Address In The Rebuttal:**

Is the deep Q-network weight initialization kept random initially?

Besides target landmark-based localization, can this method apply to feature/intensity-based object localization?

It is not clear how the initial number of clusters for k-means is chosen based on the reward distribution obtained from the previous task to continually learn the current task. Should the number of clusters always be equal to the compression ratio?

---

### Official Review · Reviewer_y9LZ · 2023-02-05

**Confidence:** 3
**Preliminary Rating:** 4

**Summary:**

Authors present a novel way to generate experience replay buffers for continual reinforcement learning that leads to substantially smaller buffers without sacrificing from the performance too much. The main principle is to sample ERBs of a given run based on the distribution of the reward values. To that end, they use a clustering algorithm to generate a small number of clusters, pick the ERB sample closest to the center and assign a weight to this sample based on the number of samples belonging to the cluster. During a retraining round, these samples are replayed based on the weight. Experiments with locating ventricle in BRATS dataset show promising results.

**Strengths:**

- Reinforcement learning for medical imaging is a very promising avenue. Here authors address a serious shortcoming of such methods, their adaptability to changing imaging environments.
- The proposed technique is simple yet effective. I can see how the model will preserve the distribution of the reward values.
- Experimental results are very promising. Even with 40x compression, the model is able to retain its performance to some extent. With 10x, the performance is almost completely retained.
- The article is really well written.

**Weaknesses:**

- The underlying assumption that the joint distribution of state-action-reward-next state can be represented only with the reward distribution is a big jump. I wish authors discuss this assumption and the resulting limitations further.
- The choice of the experiment is a bit odd and not a particular case where RL shines. I wish authors experimented with applications where RL really shines, e.g., landmark detection in whole body MRI. This reduces my enthusiasm a bit.

**Deanonymize Review:**

no

**Paper Type:**

methodological development

**Questions To Address In The Rebuttal:**

- Discussion on the main assumption would be very useful.
- A discussion of the oscillatory behavior visible in Figure 5 would be great.
- If possible, results on other datasets would help support the development here.
- It is unclear to me why authors used a clustering method rather than directly sampling from the reward distribution (through inverse cdf sampling for instance). Is there a reason?

---

### Official Review · Reviewer_5HTA · 2023-02-06

**Confidence:** 4
**Preliminary Rating:** 2

**Summary:**

The authors present a methodological development in the area of RL for medical imaging. Specifically, the paper presents a way of avoiding catastrophic forgetting of medical imaging models by compressing the experience replay buffer using coresets.

The paper is relatively clear and easy to read. The authors test the method on the BRATS dataset with different levels of compression. They find that the errors increase as the compression ratios are increased, with the 20x, 30x and 40x compression ratios being more degraded than the 10x.



**Strengths:**

The paper aims to solve an important challenge, that is of particular relevance in medical datasets: lifelong learning by avoiding catastrophic forgetting of previous data. This is common in RL tasks.

The paper is clear and suitably structured. It provides an overview of the existing literature that is mostly satisfactory (see weaknesses for further comments on this).

The presented idea is intuitive and the experimental setup is mostly consistent with the objective, although the details are lacking (see weaknesses).

**Weaknesses:**

- Although the literature review is satisfactory for the medical imaging part, it makes no reference and comparison to existing coreset-based RL replay experience buffer methods, here I note [1] and [2]. Mentioning this prior work and discussing the differences to them is crucial for placing this new work properly in the literature

- The authors perform statistical significance tests but I believe they are using an incorrect testing regime. They aim to show that the compressed version of the replay buffer performs as well as the uncompressed version. They do this by testing if the null-hypothesis should be rejected using a Wilcoxon signed-rank test. However, rejecting the null-hypothesis is not equal to accepting the null-hypothesis. Instead, it would be proper to test if the uncompressed and the compressed are from the same distribution.

- Related to the previous point, the authors perform multiple tests but do not adjust the threshold for multiple testing. If the authors do not feel multiple testing adjustment applies here, it would be good to add the motivation.


[1] Lazic, N., Yin, D., Abbasi-Yadkori, Y. and Szepesvari, C., 2021, July. Improved regret bound and experience replay in regularized policy iteration. In International Conference on Machine Learning (pp. 6032-6042). PMLR.

[2] Tiwari, R., Killamsetty, K., Iyer, R. and Shenoy, P., 2022. Gcr: Gradient coreset based replay buffer selection for continual learning. In Proceedings of the IEEE/CVF Conference on Computer Vision and Pattern Recognition (pp. 99-108).

**Deanonymize Review:**

no

**Detailed Comments:**

- Wilkoxon -> Wilcoxon
- Figure 5 has smoothing applied to it. It would be much clearer to plot the datapoints directly with no lines (or straight lines), if smoothed, it would be better to get the function for the smoothing model
- Why are the p values for 30x and 40x compression given as ranges? Why not the exact values?

**Paper Type:**

methodological development

**Questions To Address In The Rebuttal:**

- Improved intro/discussion on previous methods using coresets for ERM compression
- Better testing for statistical significance. Test for similarity, not for difference. Clarity on how adjustments for multiple tests are made.
- Added discussion on the practicalities of the work. The dataset of 100 MRIs and 24 tasks comes down to 80 MB of savings. What is the relationship between sample sizes and tasks with EMB size? Is it linear for both? In a clinical setting with thousands of MRIs and a hundred tasks, what would the savings be?

---

### Meta-Review · Area_Chair_ufmL · 2023-02-22

**Recommendation:** Accept (Poster)
**Confidence:** 4

**Metareview:**

This paper presents an approach based on reinforcement learning (RL) to address the problem of lifelong learning in medical imaging, with an application to object localization in MRI images.
All reviewers point out the value of this paper, the fact that it is well written, and its relatively novel contribution, as well as the importance of addressing avoiding catastrophic forgetting.

One reviewer pointed out possible weakness in the statistical tests used, which were explained and corrected by the authors and reported in an updated verison of the manuscript.
Similarly, most concerns from other reviewers were addressed by the authors.
However, that authors posted their replies and update their paper in the last hours of the last day of the rebuttal period (the system indicates February 18), which probably did not give reviewers time to address them and update their score. The initial evaluation was overall fairly positive.